# Proposal for Endoscopic Ultrasonography Classification for Small Pancreatic Cancer

**DOI:** 10.3390/diagnostics9010015

**Published:** 2019-01-23

**Authors:** Shuzo Terada, Masataka Kikuyama, Shinya Kawaguchi, Hideyuki Kanemoto, Yoshihiro Yokoi, Terumi Kamisawa, Sawako Kuruma, Kazuro Chiba, Goro Honda, Shinichiro Horiguchi, Jun Nakahodo

**Affiliations:** 1Department of Gastroenterology, Shizuoka General Hospital, Shizuoka 420-8527, Japan; m01060st@jichi.ac.jp (S.T.); shinya-kawaguchi@i.shizuoka-pho.jp (S.K.); 2Department of Gastroenterology, Tokyo Metropolitan Cancer and Infectious Diseases Center Komagome Hospital, Tokyo 113-0021, Japan; kamisawa@cick.jp (T.K.); sawako@cick.jp (S.K.); kazuro_oruzak@yahoo.co.jp (K.C.); 3Department of Surgery, Shizuoka General Hospital, Shizuoka 420-8527, Japan; kanemot@gmail.com; 4Department of Surgery, Shinshiro Municipal Hospital, Aichi 441-1387, Japan; y.yokoi@shinshirohp.jp; 5Department of Surgery, Tokyo Metropolitan Cancer and Infectious Disease Center Komagome Hospital, Tokyo 113-0021, Japan; ghon@cick.jp; 6Department of Pathology, Tokyo Metropolitan Cancer and Infectious Disease Center Komagome Hospital, Tokyo 113-0021, Japan; s.horiguchi@cick.jp; 7Department of Human Pathology, Juntendo University, Tokyo 113-8421, Japan; nakajun58@yahoo.co.jp

**Keywords:** pancreatic cancer, small pancreatic cancer, early diagnosis, carcinoma in situ (CIS), endoscopic ultrasonography (EUS), serial pancreatic-juice aspiration cytologic examination, SPACE

## Abstract

Backgrounds: Endoscopic ultrasonography (EUS) is used to observe the stricture of the main pancreatic duct (MPD) and in diagnosing pancreatic cancer (PC). We investigate the findings on EUS by referring to the histopathological findings of resected specimens. Materials and Methods: Six patients with carcinoma in situ (CIS) and 30 patients with invasive carcinoma of 20 mm or less were included. The preoperative EUS findings were classified as follows. A1: Simple stricture type—no findings around the stricture; A2: Hypoecho stricture type—localized hypoechoic area without demarcation around the stricture; A3: Tumor stricture type—tumor on the stricture; B: Dilation type—the dilation of the pancreatic duct without a downstream stricture; C: Parenchymal tumor type—tumor located apart from the MPD. Results: Classes A1 and A2 consisted of 2 CISs, and 4 invasive carcinomas included two cases smaller than 5 mm in diameter. Most of the cancers classified as A3 or C were of invasive carcinoma larger than 5 mm in diameter. All cancers classified as B involved CIS. Serial pancreatic-juice aspiration cytologic examination (SPACE) was selected for all types of cases, with a sensitivity of 92.0%, while EUS-guided fine needle aspiration cytology (EUS-FNA) was only useful for invasive carcinoma, and its sensitivity was 66.7%. Conclusions: Stricture without a tumor could be a finding for invasive PC and pancreatic duct dilation without a downstream stricture could be a finding indicative of CIS. Carcinoma smaller than 5 mm in diameter could not be recognized by EUS. SPACE had a high sensitivity for diagnosing small PC.

## 1. Introduction

Pancreatic cancer (PC) has a poor prognosis, mainly due to delayed diagnosis [1]. Various factors are related to delayed diagnosis, including a poor understanding of symptoms and high-risk factors suggesting PC development, or the use of conventional imaging methods for the diagnosis including contrast-enhanced computed tomography (CE-CT) and magnetic resonance imaging (MRI), which have less sensitivity and, therefore, cannot always detect small PC [2,3,4].

As an alternative to these methods, endoscopic ultrasonography (EUS) has gradually become accepted as a means of diagnosing small PC, because it is more sensitive and accurate than CE-CT or MRI [5]. Value is added to the diagnosis through the use of EUS-guided fine needle aspiration cytology (EUS-FNA) [6]. Additionally, diagnosis of carcinoma in situ (CIS) has been improved by the development of a new endoscopic retrograde cholangiopancreatography (ERCP)-related procedure of pancreatic juice cytology using nasopancreatic tube placement (serial pancreatic aspiration cytologic examination (SPACE)) [7,8,9]. Recently, the number of reports of CIS and small PC diagnosed by these methods has increased [10].

For diagnosing small PC, recognizing dilatation of a main pancreatic duct (MPD) or a cystic lesion on CT or MRI is important [3]. These findings could be secondary, due to MPD stricture by PC. Ordinarily, we see various kinds of pancreatic duct changes, including MPD stricture or dilation. Some of these are associated with PC [11,12]. EUS mostly contributes to observing the stricture and diagnosing PC, because it provides us with the highest sensitivity for PC. However, small PC, especially CIS, is rarely reported, and the key features of strictures found on EUS have not been fully researched.

In this article, we investigate and classify the findings on EUS by referring to the histopathological findings of resected specimens in patients undergoing surgery for small pancreatic cancer, including CIS.

## 2. Materials and Methods 

From January 2011 to December 2017, 217 patients underwent an elective pancreatic resection, with a final histological diagnosis of PC, at Shizuoka general hospital, Shinshiro Municipal Hospital. Patients with intraductal papillary mucinous carcinoma were not included. Among them, 40 patients had small pancreatic cancer. Small pancreatic cancer was defined by CIS and an invasive carcinoma of 20 mm or less (pT1). Because four patients with pT1 were excluded due to absence of preoperative EUS, 36 were included in the study. According to the classification of UICC 8th edition [13], we defined invasive cancer with diameter of 5 mm or less, 6 to 10 mm, and 11 to 20 mm as pT1a, pT1b, and pT1c, respectively. 

For the preoperative examination by EUS, we mainly used GF UE-260 (Olympus, Tokyo, Japan) for endoscopy and EU-ME1 (Olympus, Tokyo, Japan) as an observation apparatus. In a small number of cases, GF UM-240 (Olympus, Tokyo, Japan) or EU-ME2 (Olympus, Tokyo, Japan) was used. The study is approved from Shizuoka General Hospital Institutional Review Board. Approval number is SGHIRB#2017024, and date is 30 March 2018.

In this study, the preoperative EUS findings of changes in the pancreatic duct by small PC were classified as follows, and compared with the pathological findings of the resected specimen. We also examined preoperative diagnostic procedures for PC in the 36 patients.

## 3. Classification of EUS findings

The EUS findings were divided into two groups according to whether they had a stricture of the MPD or not (Figure 1 and Figure 2). Findings of a stricture of the MPD were classified into three types defined as follows: A1: Simple stricture type—an MPD was strictured without any other findings;A2: Hypoecho stricture type—a strictured part was surrounded by a focal hypoechoic area without clear demarcation;A3: Tumor stricture type—a tumor with clear demarcation was recognized in the strictured part.

Findings other than pancreatic duct stricture were classified to groups B and C:B: Dilation type—an MPD or a branch duct was dilated without a downstream stricture;C: Parenchymal tumor type—a pancreatic tumor with clear demarcation located apart from the MPD.

In this classification, a “stricture” was defined as an apparent change of the diameter of the MPD. That is, the diameter of the MPD becomes significantly smaller than that of the downstream MPD.

## 4. Results

### 4.1. Patient Characteristics

Seventeen males and 19 females were included. The average age was 70.7 years (56 to 84). The pancreas head, body, and tail were affected by cancer in 18, 14, and 4 patients, respectively. According to T factor, based on UICC 8th edition, 6, 6, 7, and 17 patients were classified into Tis, T1a, T1b, and T1c, respectively. 

### 4.2. EUS Findings and Pathological Findings

The EUS findings were compared with the histopathological findings of the resected specimens (Figure 3).

A1: Simple stricture type. Four patients were included. Three patients had invasive carcinoma in the pancreatic head. In these patients, severely dysplastic epithelium had widely spread in the main and branch pancreatic ducts, a part of which had changed due to invasive carcinoma with fibrous stroma. The diameters of the invasive parts were 3 mm (Figure 4), 4 mm, and 12 mm, respectively. Regional lymph node metastasis was observed in the patient with invasion of 4 mm. The remaining patients had CIS in the pancreatic tail with surrounding fibrous stroma, and focal fat replacement of the parenchyma. 

A2: Hypoecho stricture type. Two patients were included. One patient had CIS in the pancreatic tail, surrounded by fibrous stroma accompanied by focal fat replacement (Figure 5). Another one had invasive carcinoma in the pancreas body affecting the MPD with the diameter of 18 mm (pT1c). Adjacent to the carcinoma, focal fat replacement was recognized.

A3: Tumor stricture type. Twenty patients were included. The head and the body were affected in 11 and 8 patients, respectively, and the tail was affected in a single patient. In 19 patients (95%), invasive carcinoma with fibrous stroma affecting the MPD was revealed, histopathologically (Figure 6). Four and 15 tumors were classified as T1b and T1c, respectively. The remaining patients (5%) had CIS in the pancreatic head with abundant fibrous stroma surrounding the MPD forming a mass. This feature was a so-called mass forming pancreatitis (Figure 7).

B: Dilation type. Three patients were included. All patients had CIS; the affected pancreatic duct, including the MPD, and the branch were dilated to various degrees. One patient had CIS in the pancreas head, widely spreading on the accessory pancreatic duct and branches, with mild dilation. This patient had a concomitant branch-duct type intraductal papillary mucinous neoplasm (IPMN) in the pancreas body. In another two patients, CIS had widely spread on the main and branch pancreatic ducts of the pancreatic body. The affected duct was dilated, with extreme atrophy of the surrounding parenchyma, with fat replacement, without any downstream stenosis of the MPD (Figure 8).

C: Parenchymal tumor type. Seven patients were included. In three patients, carcinoma was located in the head, and in another three, it was located in the body. In the remaining patients, it was located in the tail. Three and four carcinomas were classified into pT1b and pT1c, respectively. All carcinomas were invasive with abundant fibrous stroma (Figure 9). Tumors were distant from the MPD, while, in three patients, dysplastic epithelium continued to the MPD, on the branch duct. 

In summary, the relationship between the EUS findings and the histopathological finding of the resected specimens is that A1 and A2, without an obvious tumor, involved CIS and invasive carcinoma. Two of the three invasive carcinomas were of stage T1a. On the other hand, A3 and C, with detectable tumors, were of stages T1b and T1c. That is, the tumors recognized on EUS were larger than 5 mm. Every carcinoma classified as B, showing pancreatic duct dilation without a downstream stricture, involved CIS. 

### 4.3. Preoperative Diagnostic Procedures 

All nine patients in categories A1, A2, and B, without an apparent tumor on EUS, underwent a serial pancreatic-juice aspiration cytologic examination (SPACE). This resulted in positive results for cancer in eight patients (88.9%) (Figure 10). One patient, with a negative result for cancer by SPACE, belonged to category B, and selected surgical treatment without preoperative confirmation of diagnosis of cancer. 

Among the 27 patients in the A3 (20) and C (7) categories with a recognized tumor, 11 and 13 underwent EUS-FNA and SPACE for preoperative diagnosis for PC, with positive results in seven (63.6%) and 12 (92.3%), respectively (Figure 10). Three among the four patients with a negative result by EUS-FNA, belonging to A3 (including a case of CIS with so-called mass forming pancreatitis), alternatively underwent SPACE with a positive result for PC. One patient with a negative result by SPACE, belonging to A3, alternatively underwent EUS-FNA with a positive result for PC. Four patients, including three without selecting preoperative diagnostic procedures and one with a negative result for PC by EUS-FNA underwent surgical treatment without preoperative confirmative diagnosis for PC. 

Finally, SPACE results were positive for cancer in 23 of 25 sessions (92.0%), while EUS-FNA was positive in eight of 12 sessions (66.7%). The sensitivity of SPACE was higher than that of EUS-FNA, but there was no statistically significant difference (*p* = 0.073). Four patients selected surgery without a confirmative preoperative diagnosis. For small pancreatic cancer, SPACE is frequently selected to confirm diagnosis with a high diagnostic ratio.

## 5. Discussion

Among PC, CIS is rare and pT1 cancer is not common. In our study, six (2.8%) cases of CIS and 34 (15.7%) cases of invasive cancer of 20 mm or less (pT1) among 217 surgically resected pancreatic cancers were included. In a previous report—a multicenter study on the early diagnosis of pancreatic cancer in Japan [10]—the incidence of CIS was 1.6% among surgically resected pancreatic cancers. Another study on pancreatic cancer reported that the incidence of pT1 was 11.3–12.5% [15]. Although early-stage pancreatic cancer could be diagnosed by performing EUS-FNA or SPACE in patients with clinical manifestations, suggesting pancreatic cancer [8,16], the imaging features of an early-stage pancreatic cancer have not been fully investigated. This study aims to clarify and classify EUS findings of small PC, and compare the findings with the histopathological ones of the resected specimens. This is with the aim of creating a foundation for diagnosing small pancreatic cancer using EUS, by considering the histopathological background of the EUS findings. 

The ductal change of stricture is an important sign of pancreatic cancer. The hypoecho stricture type (A2) has been described as a typical finding of CIS [17]. However, our study showed that any type of stricture could not only be a sign of CIS, but also a sign of invasive carcinoma. We found that three patients belonging to simple stricture type (A1) had invasive carcinoma. A previous report [18] described a case with simple stricture type advanced to invasive carcinoma with lymph node metastasis for three months. That is, regardless of the surrounding finding, any type of stricture could be a sign of PC and become an indicator for a further examination, such as EUS or SPACE.

This is also found regarding invasive carcinoma: CIS causes stricture of the MPD by surrounding fibrosis. Fibrosis is responsible for the hypoechoic area at the strictured part on EUS, recognized as a hypoecho stricture type (A2). The cause of fibrosis is presumed to be focal pancreatitis, whereas cytokines, such as TGF-β, from cancer cells could cause fibrosis, as preparation for cancer cells to invade surrounding tissue [19]. On the other hand, there are cases without the hypoechoic area, even when the stricture was surrounded by fibrosis [14,20]. In our study, three cases of invasive carcinoma, classified as simple stricture type (A1) in the pancreatic head, were accompanied with fibrosis, but no fatty change of the parenchyma was recognized around the stricture. No fatty change around the stricture might lead to the absence of a hypoechoic area. Not only fibrosis, but also fat change, could contribute to the hypoechoic area at the section of stricture [14].

Tumor stricture type (A3) is a typical finding of PC. Within the A3 category, a tumor is revealed by a focal and hypoechoic area with well-demarcation on an EUS. On the other hand, CIS could also form a tumor-like lesion [21]. One of the A3 patients had CIS with a tumor-like appearance that consisted of abundant fibrosis, inflammatory cell infiltration, and the so-called mass-forming pancreatitis. This can be caused by a case of autoimmune pancreatitis (AIP) with CIS [22]. However, our case was not accompanied by AIP.

Among our cases of CIS, three revealed dilation of the pancreatic duct without a downstream stricture [11]. These cases are classified into dilation type (B). Two cases of the pancreas body had severe parenchymal atrophy, with fat replacement around the dilated pancreatic duct, and CIS. Usually, pancreatic dilation is believed to be caused by pancreatic juice flow obstruction due to pancreatic duct stricture, but these cases did not have a downstream stricture of the pancreatic duct. An IPMN could not be found either. The intraductal pressure of the pancreatic duct is 16.2 ± 8.7 mmHg [23], which is higher than that of the intraperitoneal pressure of 5–7 mmHg. The gradient of pressures could make the pancreatic duct dilate when the pancreatic parenchyma becomes atrophic. The last case of CIS in the pancreas head did not have pancreatic parenchymal atrophy. The CIS spread widely on the branch duct of the accessory pancreatic duct area, and the branch duct slightly dilated. This case had a branch-duct type IPMN in the pancreas body, and an IPMN might influence the pancreatic duct dilation. 

Tumors classified to type C, located in the pancreatic parenchyma, did not cause MPD stricture. In this type, indirect findings of the pancreatic duct could not be expected to contribute to the diagnosis of PC, and recognizing a tumor itself was the only way of diagnosing PC. In general, US, contrast-enhanced CT, and MRI are used for diagnosing pancreatic cancer, but several reports describe that they do not fulfill the purpose. US has both low detective sensitivity and specificity [2], contrast-enhanced CT has insufficient detective sensitivity for PC of 20 mm or less [3], and MRIs detect cystic lesions well, but their imaging of small solid tumors is insufficient [4]. Compared with these methods, EUS is superior for diagnosing PC [16] [24], and EUS should be selected in the patients having clinical manifestations or high-risk factors for PC [16]. However, our result shows that A3 and C, with a detectable tumor, consisted of T1b and T1c. That is, the tumors recognized on EUS were larger than 5 mm. Diagnosing cancer based on the size of T1a remains challenging despite the use of EUS.

PC originates from the pancreatic ductal epithelium. When PC is diagnosed as CIS, the tumor stage is 0, and this is known as very early PC. PC with stage 0 is very rare, and the number of cases reported in Japan is only 50 to date. However, diagnosis is possible, and the method of confirmation is SPACE. SPACE is reported to have high sensitivity for PC [8,16], and has contributed to the diagnosis of CIS in our study in patients with pancreatic stricture, including those with negative results on EUS-FNA. However, post-ERCP pancreatitis, after placing a nasopancreatic tube, should be of concern [25].

Any pancreatic duct stricture on any kind of imaging method, such as US, CT, or MRI, suggests PC, and EUS should be performed to observe the stricture. If a tumor is recognized well on EUS, EUS-FNA could be the first-choice diagnostic method. However, our results suggest that SPACE could make a superior contribution, and should be selected to confirm the diagnosis in small PC. However, this study has a limitation because it is retrospective and is not a case–control study.

## 6. Conclusions

EUS could provide a precise observation of MPD and pancreas parenchyma. Understanding the association between the EUS findings and the histopathological findings could allow us to diagnose PC at an earlier stage. Our investigation suggests that a simple stricture without a tumor or a hypoechoic area could be a sign of invasive PC, and that pancreatic duct dilation without a downstream stricture could be a sign of a CIS. Carcinomas with a diameter smaller than 5 mm could not be recognized by EUS. Moreover, SPACE is more effective than EUS-FNA for diagnosing small PC. 

## Figures and Tables

**Figure 1 diagnostics-09-00015-f001:**
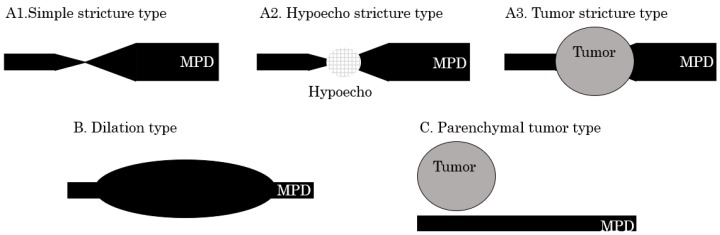
Classification of endoscopic ultrasonography (EUS) findings. **A1**: Simple stricture type—a main pancreatic duct (MPD) is strictured without any other findings; **A2**: Hypoecho stricture type—a strictured part is surrounded by a focal hypoechoic area without clear demarcation; **A3**: Tumor stricture type—a tumor with clear demarcation is recognized at the strictured part; **B**: Dilation type—An MPD or a branch duct is dilated without a downstream stricture; **C**: Parenchymal tumor type—A pancreatic tumor with clear demarcation is located apart from the MPD.

**Figure 2 diagnostics-09-00015-f002:**
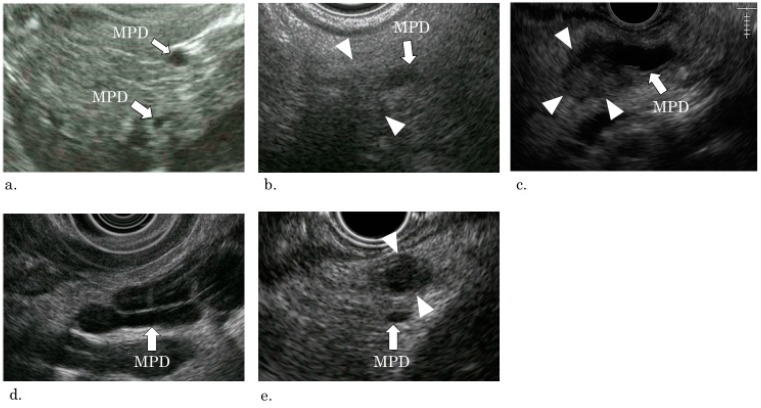
EUS pictures of each type of classification. (**a**) A1: Simple stricture type—the main pancreatic duct is strictured with a slight dilation of the upstream pancreatic duct, but no other finding (such as hypoecho or a tumor) is recognized at the strictured part (arrow, main pancreatic duct). (**b**) A2: Hypoecho stricture type—the main pancreatic duct is strictured by a blurred hypoechoic lesion with slight dilation of the upstream pancreatic duct (arrow, main pancreatic duct; arrow head, hypoechoic lesion). (**c**) A3: Tumor stricture type—A tumor strictures the main pancreatic duct with upstream dilation (arrow, main pancreatic duct; arrow head, tumor). (**d**) B: Dilation type—the main pancreatic duct and the branch duct is dilated without a tumor or a downstream stricture (arrow: main pancreatic duct). The cystic dilated duct is a branch duct. (**e**) C: Parenchymal tumor type—A tumor exists distant from the main pancreatic duct. (arrow, main pancreatic duct; arrow head, tumor).

**Figure 3 diagnostics-09-00015-f003:**
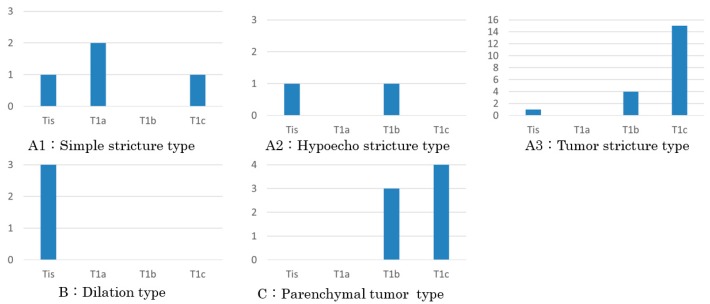
T category of the lesions. Categories (**A1**) and (**A2**) without an obvious tumor include carcinoma in situ (CIS) and invasive carcinoma. Two of the three invasive carcinomas of A1 were T1a. (**A3**) and (**C**) with a detectable tumor almost consist of T1b and T1c. Every carcinoma of (**B**) showing pancreatic duct dilation without a downstream stricture is CIS.

**Figure 4 diagnostics-09-00015-f004:**
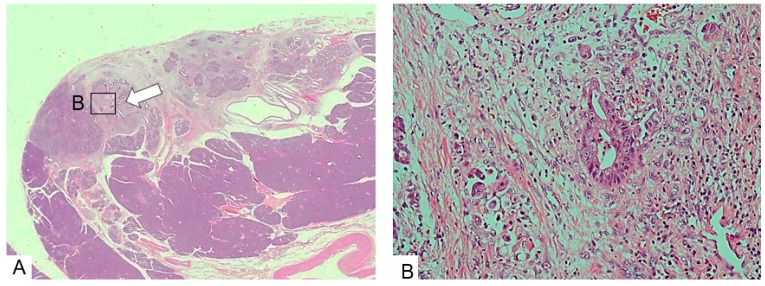
Simple stricture type. (**A**) The main pancreatic duct (arrow) is strictured and surrounded by dense fibrous stroma. (**B**) The epithelium changes to a dysplastic one with an invasive part within a minimal area of 3 mm. (Hematoxylin and eosin stain (H.E), ×100). Copyright information © Japan Pancreas Society 2017 [14].

**Figure 5 diagnostics-09-00015-f005:**
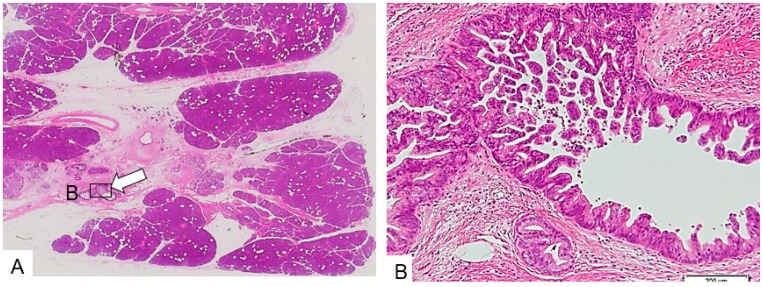
Hypoecho stricture type. (**A**) Histopathological specimen does not show the severely collapsed main pancreatic duct (arrow). Surrounding parenchyma becomes atrophic and is partially replaced by fat tissue. The main and the branch pancreatic duct with epithelial papillary growth is surrounded by fibrous stroma. (**B**) High.power microscopic view reveals the papillary growing epithelium as CIS. (H.E., ×100). Copyright information © Japanese Society of Gastroenterology 2017 [9].

**Figure 6 diagnostics-09-00015-f006:**
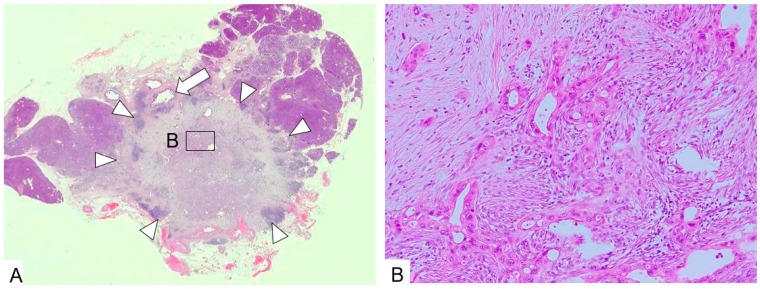
Tumor stricture type (1). (**A**). A tumor with an irregular margin (arrow heads) occupies the pancreatic parenchyma. The main pancreatic duct cannot be identified. A dilated duct (arrow) is the branch pancreatic duct. (**B**). The tumor consists of dense fibrous tissue with small irregular glands. The epithelium of the gland is severely dysplastic and the gland is a so-called cancer pseudogland. (H.E., ×100).

**Figure 7 diagnostics-09-00015-f007:**
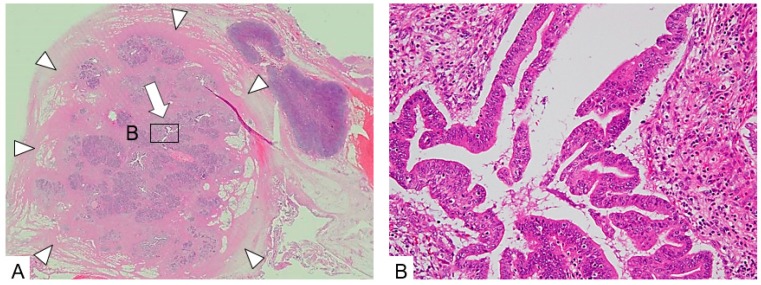
Tumor stricture type (2). (**A**) A large tumorous lesion with an irregular margin (arrow heads) is seen with a collapsed pancreatic duct (arrow) at the center of the lesion. The lesion consists of atrophied acinar with rich fibrous stroma. (**B**) The ductal epithelium changes to CIS surrounded by inflammatory cell infiltration and fibrous tissue. The diagnosis is so-called mass-forming pancreatitis with CIS. (H.E., ×100).

**Figure 8 diagnostics-09-00015-f008:**
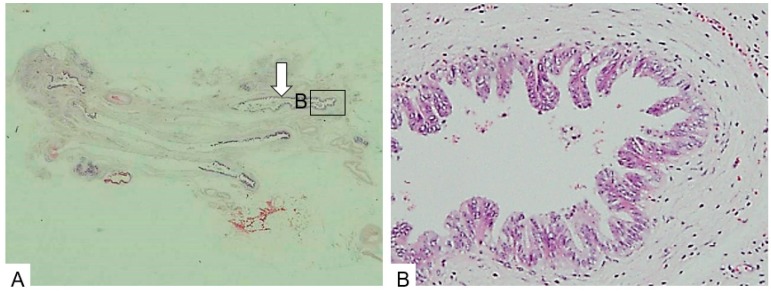
Dilation type. (**A**). The pancreatic parenchyma is severely atrophic and replaced by fat. Ducts are dilated (arrow). (**B**). The ductal epithelium shows CIS. (H.E., ×100). Copyright information © Japan Pancreas Society 2015 [11].

**Figure 9 diagnostics-09-00015-f009:**
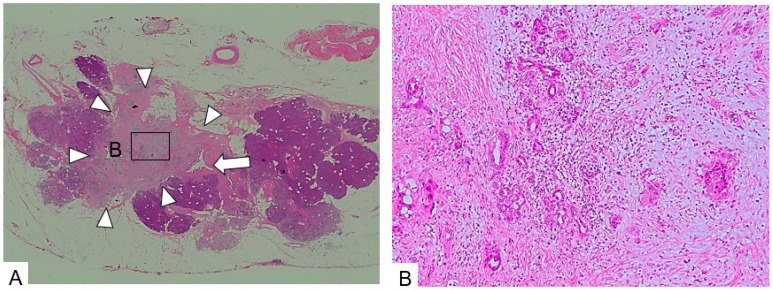
Parenchymal tumor type. (**A**). A tumor with an irregular margin is recognized (arrow heads) without invading the main pancreatic duct (arrow). The main pancreatic duct has a fibrously thickened wall, which is in contact with the tumor, but has not been invaded. (**B**). The tumor consists of dense fibrous stroma with cancer pseudoglands. (H.E., ×100).

**Figure 10 diagnostics-09-00015-f010:**
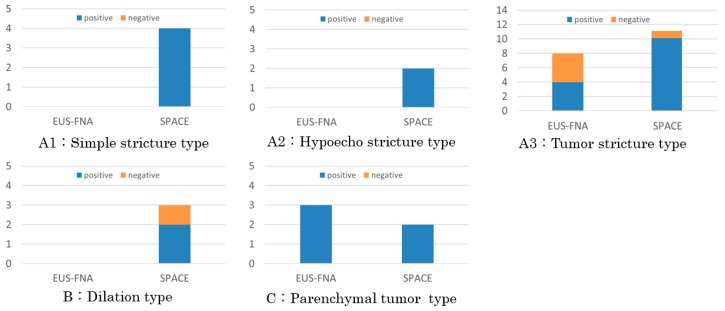
Preoperative diagnostic procedures. Serial pancreatic-juice aspiration cytologic examination (SPACE) was selected for not only patients in categories (**A1**), (**A2**), and (**B**), without an apparent tumor, but also in categories (**A3**), and (**C**), with a recognized tumor, and sensitivity of SPACE was 92.0%, while EUS-guided fine needle aspiration cytology (EUS-FNA) was used for patient in categories (A3), and (C), and sensitivity was 66.7%.

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
