# Peer review of "Proposal for Endoscopic Ultrasonography Classification for Small Pancreatic Cancer"

_diagnostics, 2019, doi:10.3390/diagnostics9010015_

Round 1
Reviewer 1 Report
In this study, the authors aim to classify endonosonographic findings of small pancreatic cancers and correlate with histopathology. The key issue is that the authors used a specified patient population, i.e., resected pancreatic cancers. There is no control arm in the study. Hence the results are not generalizable.
If the study was construed in a way to include all comers including those with chronic pancreatitis, and other pancreatic pathology, then the relevance of this classification and the diagnostic accuracies would be more relevant. A lack of a control arm would need to mentioned as study limitations.
Nevertheless, this study is well conducted where the authors perform a comprehensive analysis of EUS characteristics of small pancreatic cancers. This would be very useful to endosonographers in the appropriate clinical setting, i.e, while evaluating a patient with a subtle stricture or dilation and in the absence of other obvious pancreatic pathology such as chronic pancreatitis.
Author Response
Thanks you for reviewing our paper out of your busy schedule. As you pointed out, benign diseases such as chronic pancreatitis sometimes cause tumor and pancreatic duct change which are difficult to distinguish from pancreatic cancer. We assume that prospective case control study is next issue for proving the usefulness of EUS classification and SPACE. We add the sentence “However, this study has a limitation because it is retrospective and not a case-control study.” as per your instructions.
Reviewer 2 Report
Here the authors assess the classification of Endoscopic ultrasonography (EUS) for small pancreatic cancer by referring to the histopathology in the resected specimens. This is great work, however, the authors should pay more attention to the figures, some suggestions regarding the manuscript style are provided as below:
1. All the figures are too big in size, make them suitable, do not leave so much blank.
2.Table 1 is not necessary, this content could be explained in the methods.
3. Again, think about the figures, the current figures are in a mess, please combine some to one panel instead of one figure.
Author Response
Thanks you for reviewing our paper out of your busy schedule.
1. All the figures are too big in size, make them suitable, do not leave so much blank.
→We deleted the blank in the figure and reduced the size of the figure.
2.Table 1 is not necessary, this content could be explained in the methods.
→We deleted Table1.
3. Again, think about the figures, the current figures are in a mess, please combine some to one panel instead of one figure.
→As you pointed out, figures were in a mess. We deleted some figures and simplified it. We deleted Figure3 and12, low power field images of pathological findings.